# Sluggish Cognitive Tempo among Iranian Children and Adolescents: A Validation Study of the Farsi Child and Adolescent Behavior Inventory (CABI)–Parent Version

**DOI:** 10.3390/jcm11216346

**Published:** 2022-10-27

**Authors:** Dena Sadeghi-Bahmani, Youkhabeh Mohammadian, Mohammad Ghasemi, Laleh Sadeghi Bahmani, Nahid Piri, Annette Beatrix Brühl, Stephen P. Becker, G. Leonard Burns, Serge Brand

**Affiliations:** 1Department of Psychology, Stanford University, Stanford, CA 94305, USA; 2Sleep Disorders Research Center, Kermanshah University of Medical Sciences, Kermanshah 67158-47141, Iran; 3Department of Clinical Psychology, Kermanshah University of Medical Sciences, Kermanshah 67158-47141, Iran; 4Department of Education and Psychology, Shahid Ashrafi Esfahani University, Esfahan 81799-49999, Iran; 5Psychiatric Clinics (UPK), Center of Affective, Stress and Sleep Disorders (ZASS), University of Basel, 4002 Basel, Switzerland; 6Division of Behavioral Medicine and Clinical Psychology, Cincinnati Children’s Hospital Medical Center, Cincinnati, OH 45229, USA; 7Department of Pediatrics, University of Cincinnati College of Medicine, Cincinnati, OH 45267, USA; 8Department of Psychology, Washington State University, Washington, DC 99164, USA; 9Division of Sport Science and Psychosocial Health, Department of Sport, Exercise and Health, Faculty of Medicine, University of Basel, 4002 Basel, Switzerland; 10Abuse Prevention Research Center, Kermanshah University of Medical Sciences, Kermanshah 67158-47141, Iran; 11School of Medicine, Tehran University of Medical Sciences, Tehran 14166-34793, Iran

**Keywords:** sluggish cognitive tempo, cognitive disengagement syndrome, attention-deficit/hyperactivity disorder, validation, child behavior, adolescent behavior, Farsi

## Abstract

Background: Sluggish cognitive tempo (SCT), recently renamed cognitive disengagement syndrome (CDS), is a set of behavioral symptoms that includes excessive daydreaming, mental confusion and fogginess, being lost in one’s thoughts, and slowed behavior and thinking. Symptoms of SCT show overlap with a broad range of other symptoms such as attention-deficit/hyperactivity disorder inattention, anxiety, and depression, or oppositional defiant disorder (ODD). To measure SCT, one of the optimal measures is the Child and Adolescent Behavior Inventory (CABI). Here, we report the psychometric properties of the Farsi version of the CABI Parent Version, including the CABI SCT scale and its subscales. Methods: The participants were the parents of 209 children and adolescents (53.9% girls; ages 8–19 years; M_age_ = 14.23, SD_age_ = 2.72). Parents completed the SCT, ADHD inattention (ADHD-IN), ADHD-hyperactivity/impulsivity (ADHD-HI), oppositional defiant disorder (ODD), limited prosocial emotions (callous-unemotional (CU) traits), anxiety, depression, social impairment, and academic impairment scales of the Child and Adolescent Behavior Inventory (CABI). Parents also completed four dimensions of the Strengths and Difficulties Questionnaire (SDQ: emotional problems; conduct problems; peer problems; prosocial behavior), and five dimensions of the Kidscreen questionnaire (physical health; psychological well-being; autonomy and parental well-being; peers and social support; school environment). Results: SCT symptoms demonstrated strong discriminant validity from the ADHD-IN symptoms. SCT showed stronger first-order and unique associations than ADHD-IN with anxiety, depression, and ODD, whereas ADHD-IN showed stronger first-order and unique associations than SCT with ADHD-HI, CU, and social and academic impairment. Further, SCT showed stronger first-order and unique associations than ADHD-IN with more emotional problems, peer problems, and with lower prosocial behavior, as assessed with the SDQ. Higher scores for SCT were associated with lower psychological well-being, autonomy and parental relations, and lower peer and social support, as assessed with the Kidscreen. Higher ADHD-IN scores were associated with lower peer and social support, and a lower school environment. Conclusions: The Farsi version of the CABI–Parent Version has very good psychometric properties for assessing SCT and other dimensions of psychopathology/impairment and replicates the findings from similar studies with children and adolescents from South Korea, Spain, Turkey, and the United States. Accordingly, the present study provides further support of the transcultural validity of the sluggish cognitive tempo construct.

## 1. Introduction

Within the last two decades, the construct of sluggish cognitive tempo (SCT) has gained increasing interest among both the scientific community and practitioners caring for children and adolescents with a broad variety of attention-deficit/hyperactivity disorder (ADHD) and ADHD-like behavior [1,2,3,4]. Please note that in mid-2022, the term sluggish cognitive tempo (SCT) was replaced by the more appropriate label of Cognitive Disengagement Syndrome (CDS). However, to align with previous validation studies, in the present paper, we still use the expression sluggish cognitive tempo (SCT). SCT is characterized by cognitive disengagement and hypoactivity behaviors, including mental confusion, excessive daydreaming, lethargy, staring, slowed behavior and thinking, fogginess, underactivity, apathy and withdrawal, slowness in completing tasks, lack of initiative and efforts, and trouble staying awake and alert [2,3,5,6,7,8,9,10].

While it appears that there is the risk to over-pathologize behavioral traits that might at times be experienced by a significant proportion of children and adolescents [11], some of the benefits of identifying SCT might be in the recognition that this represents another aspect of neurodiversity. More specifically, individuals with this picture may need different adaptations in their environments, and increased understanding from families, teachers and peers, in addition to any consideration of need for medication.

For scientists, the construct of SCT is important because studies have shown that SCT is distinct from ADHD inattentive symptoms (ADHD-IN) and may not identify individuals with a “pure” ADHD inattentive presentation [2,6,12,13].

For practitioners, the concept of SCT is important to further distinguishing attention problems such as daytime sleepiness and internalizing problems from typical ADHD [14,15,16]. Importantly, children and adolescents with ADHD and co-occurring symptoms of SCT may be less likely to respond to the first-line treatment with methylphenidate [17,18]; rather, they may be more likely to respond to atomoxetine [19,20], and they may be somewhat less likely to respond to evidence-based behavioral treatment [21].

### 1.1. Findings of SCT in Children and Adolescents in the US, South Korea, Turkey, and Spain

The SCT construct is not related to a specific cultural framework, and measures to assess SCT are available in several languages. More specifically, the Child and Adolescent Behavior Inventory–Parent Version (CABI) [22] is available in American English [6], Korean [23], Turkish [5], and Spanish [24,25].

To identify children with SCT [6], 2056 US American mothers of children (mean age: 8.49 years; 49% girls) completed the Child and Adolescent Behavior Inventory (CABI [22]) [6]. Further, scores greater than the top 5% on SCT and ADHD measures were used to create subgroups of SCT-only (*n* = 53, 2.58%), ADHD-only (*n* = 93, 4.52%), SCT + ADHD (*n* = 49, 2.38%), and comparison (*n* = 1861, 90.52%) groups. Compared to children with only symptoms of ADHD, children with only SCT symptoms also reported higher scores for depression, anxiety, conflicted shyness, and sleep difficulties. In contrast, compared to children with only symptoms of SCT, children with only ADHD had greater executive function deficits and higher scores for oppositional defiant disorder (ODD). Children with SCT-only and ADHD-symptoms had similar scores for friendship, social impairment, and academic impairment.

To test the psychometric properties of the Korean version [23] of the CABI [22], 469 adolescents aged 13 to 17 years (49.8% females) and their parents completed the questionnaires. For self-report, 11 out of 15 SCT items showed convergent and discriminant validity with ADHD inattention presentation. For parent-report, all 15 SCT items showed convergent and discriminant validity with ADHD-IN. Further, SCT showed unique and stronger associations than ADHD-IN with internalizing dimensions of psychopathology. In contrast, ADHD-IN showed unique and stronger associations than SCT with externalizing dimensions of psychopathology. In addition, both SCT and ADHD-IN showed similar unique associations with social problems, whereas ADHD-IN was more strongly related than SCT to lower grades, that is to say, to lower academic performances at school. Last, irrespective from self- or parent-report, higher scores for SCT were more strongly related to internalizing psychopathologies, while higher scores for ADHD-IN were more strongly related to externalizing psychopathologies.

To test the psychometric properties of the Turkish version [5] of the CABI [22], a total of 1015 children and adolescents aged from 6 to 15 years (mean age: 10.05 years; 56% females) took part in this validation study (253 of whom were attending an out-patient psychiatric ward). Results showed excellent internal validity of the SCT scores across the samples and across gender, with SCT items clearly distinguishable to ADHD-IN. Further, SCT scores showed stronger first-order and unique associations than ADHD-IN with symptoms of anxiety and depression. In contrast, ADHD-IN scores showed stronger first-order and unique associations than SCT with ADHD hyperactivity-impulsivity (ADHD-HI) symptoms, ODD, and academic impairments. Last, both higher scores for SCT and ADHD-IN showed similar associations with callous-unemotional (CU) behavior and social impairments.

Sáez et al. [25] analyzed the psychometric properties of the Spanish version [24] of the CABI [22], as completed by mothers, fathers, and teachers among a longer sample of 2142 children and adolescents (49.49% girls; ages 8 to 13 years). Sáez et al. showed that the 15 SCT items demonstrated convergent and divergent validity of the 15 SCT symptoms with discriminant validity with ADHD-IN across all informants. Further, the SCT symptom ratings also showed measurement invariance across mothers, fathers, and teachers, and SCT and ADHD-IN factors had different and unique associations with other symptom and impairment factors.

### 1.2. The Current Study

The main aims of the current study were to translate the CABI–Parent Version [22] into Farsi, to test its psychometric characteristics, and to investigate if and to what extent SCT traits were associated with specific psychological functioning such as internalizing problems and social problems with parents and peers, compared to ADHD-IN traits, as already observed among children and adolescents in the USA [6], Spain [25], Turkey [5], and South Korea [23]. For the following reasons, we claim that present study is of practical and clinical importance. First, the Farsi version of the CABI allows to investigate whether and to what extent dimensions of SCT might be observable among children and adolescents with a Farsi-speaking background. Following Eberhard et al. [26], Farsi is the official language of Iran, Afghanistan, and Tajikistan. While it is estimated that about 77.4 million people speak Farsi as their first language, about 40 million people use Farsi as their second language [27], in addition to minorities in Uzbekistan, United States, Pakistan, Turkey, United Arab Emirates, Iraq, Qatar, Germany, India, or Canada [26]. Second, the validation of the Farsi CABI allows further studies to corroborate the prevalence and stability of SCT, irrespective from the cultural background. Third, given that children and adolescents with SCT may be mistakenly diagnosed with ADHD-IN and thus inappropriately treated, the Farsi version of the CABI might help to identify those children and adolescents in the need of a more appropriate psychopharmacological and psychotherapeutic treatment (see [19,20,21]). Of note, although this study focuses on SCT symptoms, this study further examines the psychometric support of the other CABI–Parent Version subscales, which will increase the measure’s utility in Farsi-speaking populations.

The three hypotheses and research question were as follows: First, based on previous studies [5,6,23,25] we assumed that SCT scores would form a factor, which was clearly distinguishable from the factor ADHD-IN. Second, based on the same studies, we expected that compared to ADHD-IN, SCT would strongly correlate with internalizing symptoms. Complementarily and third, we expected that compared to SCT, ADHD-IN would strongly correlate with externalizing symptoms. Previous research showed that both SCT and ADHD-IN symptoms were both associated with social impairments [5,6,23] and callous-unemotional (CU) behaviors [5], while in another study [6], ADHD symptoms only were more strongly associated with ODD symptoms. With the research question, we sought to determine whether, in the Farsi population, SCT symptom dimensions identified through use of the CABI differed in their patterns of association with other psychopathology and with impaired social behavior when compared with ADHD-IN symptom dimensions. We hold that the present results have the potential to provide a useful and psychometrically sound measure to identify children and adolescents with SCT traits among Farsi-speaking communities. We further hold that identifying children and adolescents with SCT traits misleadingly diagnosed with ADHD-IN could help both the child and adolescent to obtain a more appropriate treatment; in the same vein, distinguishing between children and adolescents with SCT traits from those with ADHD (-IN) should help the practitioner to offer a more fine-grained and effective treatment.

## 2. Methods

### 2.1. Participants

The participants were the parents of 209 children and adolescents (53.9% girls; age range: 8–19 years; M_age_ = 14.23 years, SD_age_ = 2.72). The community sample was obtained from public elementary, secondary, and high schools (2nd graders to 12th graders) in Kermanshah (Iran).

### 2.2. Procedure

In Iran, schools are gender-separated; accordingly, we have chosen both female and male schools equally, and accordingly, a female psychologist was responsible for the assessment in female schools, while a male psychologist cared for the assessment in male schools. Schools were selected following clustered sampling, and to cover all age ranges, per grade and gender, and one class per school was randomly chosen. Once a dean of a school agreed with the assessment, the dean introduced the psychologist both to teachers and students. In parallel, parents were invited to attend an introduction information meeting at school. All parents and teachers were informed about the aims of the study, the voluntary participation and the secured and anonymous data handling. Next, eligible parents interested in the study signed the written informed consent and then completed the CABI–Parent Version. All questionnaires were sealed in envelopes. Further, students older than 14 years and of parents who agreed to and completed the CABI–Parent Version, signed their written informed consent and completed the CABI–Student Version at school during the second lesson between 9 am to 10 am. While all students completed the questionnaires, exclusively, questionnaires of students with signed written informed consents were further evaluated. Once a student completed the questionnaire, this was sealed in the appropriate envelope. A research psychologist supervised the procedure and collected all sealed envelopes at the end of the lesson. Teachers completed the CABI–Teacher Version at home within a week and returned the questionnaires in a sealed envelope. Parents and teachers received a gift card. Irrespective from their participation or non-participation to the study, all students present in the classroom received a voucher for the school restaurant. Further, for students, participation or non-participation had no advantages or disadvantages on their schooling credits. The Ethical Committee of the Kermanshah University of Medical Sciences (KUMS; Kermanshah, Iran) approved the study (IR.KUMS.REC.1398.016), which was performed in accordance with the seventh and current revision of the Declaration of Helsinki. In the present study, we report data of the CABI–Parent Version.

### 2.3. Measures

#### 2.3.1. Sociodemographic Information

Parents reported on their children’s age (years), and gender at birth (female; male).

#### 2.3.2. Child and Adolescent Behavior Inventory; Translation Process

To translate the English version of the CABI into Farsi, we followed the algorithms of Brislin [28] and Beaton et al. [29]. First, two independent translators translated the English version into Farsi/Persian. Next, a third independent person compared the two translations, and, in case of differences, discussed the issues and performed the final draft. Next, two further independent translators performed the back-translation and compared the back-translated English versions with the original version. Finally, the final version mirrored the general agreement of all five researchers involved in this procedure.

Parents completed the Child and Adolescent Behavior Inventory (CABI [22]). As described elsewhere [5,22,25], the CABI measures SCT (15 symptoms), DSM-5 ADHD-IN (nine symptoms), DSM-5 ADHD-HI (nine symptoms), DSM-5 ODD (eight symptoms), callous-unemotional behaviors (four symptoms of the DSM-5 Limited Prosocial Emotions specifier), anxiety (six symptoms), depression (six symptoms), social impairment (four items for parents (quality of interactions with parents, other adults, siblings, and peers)), and academic impairment (four items: quality of homework/classwork, reading skills, arithmetic skills, and writing skills). Parents rated their children’s behavior regarding the past four weeks. The SCT, ADHD, ODD, anxiety, and depression symptoms were rated on a 6-point scale (i.e., almost never (never or about once per month), seldom (about once per week), sometimes (several times per week), often (about once per day), very often (several times per day), and almost always (many times per day)). The callous-unemotional symptoms were rated on a slightly different 6-point scale (i.e., almost never (0 to 10% of the time), seldom (11 to 20% of the time), sometimes (21 to 49% of the time), often (50 to 79% of the time), very often (80 to 89% of the time), and almost always (90 to 100% of the time)). A 7-point scale was used for the academic and social impairment items (i.e., severe difficulty, moderate difficulty, slight difficulty, average performance (average interactions) for grade level, slightly above average, moderately above average, and excellent performance (excellent interactions) for grade level). The callous-unemotional, academic impairment, and social impairment items were reverse keyed so that higher scores represent more callous-unemotional, academic impairment, and social impairment, respectively.

In the current sample, Cronbach’s alpha for the SCT, anxiety, depression, ADHD-IN, ADHD-HI, ODD, callous-unemotional behavior, social impairment, and academic impairment were 0.89, 0.77, 0.84, 0.89, 0.87, 0.86, 0.83, 0.90, and 0.92. Skewness for the nine CABI scales varied from −0.87 (academic impairment) to 2.02 (ADHD-HI) with kurtosis values varying from −1.02 (callous-unemotional behavior) to 5.84 (ADHD-HI).

#### 2.3.3. Strengths and Difficulties Questionnaire (SDQ); Parent Version

Parents completed the dimensions of emotional problems, conduct problems, peer problems, and prosocial behavior (always 5 items) of the Farsi SDQ (https://www.sdqinfo.org/py/sdqinfo/b3.py?language=Farsi, accessed on 1 June 2018). Example items are: “Nervous or clingy in new situations” (emotional problems), “often fights with other children” (conduct problems), “rather solitary, tends to play alone” (peer problems), “shares readily with other children” (prosocial behavior). Answers are given on a three-point Likert scale ranging from 0 (=not true) to 1 (=somewhat true) to 2 (=certainly true). Higher scores reflect a more pronounced dimension. Note that a higher score for prosocial behavior reflects better prosocial behavior. The Farsi version [30,31] showed satisfactory psychometric properties (Cronbach’s alphas: 0.69–0.79; in the present study: Cronbach’s alphas: 0.793 emotional problems to 0.891 (conduct problems); skewness: 0.018 (emotional problems dimension to 1.506 (peer problems); kurtosis: −0.542 (conduct problems) to 1.801 (emotional problems)).

#### 2.3.4. Quality of Life; Kidscreen; Parent Version

Parents completed the Farsi 27-item version of the Quality of Life questionnaire (Kidscreen) for children and adolescents (https://www.kidscreen.org, accessed on 1 June 2018). Dimensions were: Physical well-being (5 items): e.g., “In general, how would your child rate her/his health?”; 0 = poor; 4 = excellent. Psychological well-being (7 items): e.g., “Has your child felt that life was enjoyable?”; 0 = not at all; 4 = always. Autonomy and parent relation (7 items): e.g., “Has your child been able to talk to her/his parent(s) when she/he wanted to?”; 0 = never; 4 = always. Social support and peers (4 items): e.g., “Has your child been able to rely on her/his friends?”; 0 = never; 4 = always. School (4 items): e.g., “Has your child got on well at school?”; 0 = not at all; 4 = extremely. Higher sum scores reflect more pronounced physical well-being, psychological well-being, autonomy and parent relation, social support and peer quality relationship, well-being, and performance at school. The Farsi version [32] showed satisfactory psychometric properties (Cronbach’s alphas: 0.77, −0.89; in the present study: Cronbach’s alpha: 0.773 (physical well-being) −0.901 (psychological well-being); skewness: −1.028 (social support and peers) to 1.092 (school); kurtosis: −0.320 (autonomy and parent relation) to 3.194 school).

### 2.4. Statistical Analysis

The Mplus statistical software was used for the analyses [33]. The first analysis applied an exploratory two-factor confirmatory factor analysis model to the 15 SCT items and the nine ADHD-IN items. This analysis allowed the SCT symptoms to cross-load on the ADHD-IN factor and the ADHD-IN symptoms to cross-load on the SCT factor (Geomin rotation). This analysis is also referred to as an exploratory structural equation model [34]. This analysis treated the SCT and ADHD-IN as categorically indicated and used the Mplus WLSMV estimator.

The second series of analyses determined the correlations of the SCT and ADHD-IN factors with the other CABI scale scores (i.e., anxiety, depression, ADHD-HI, ODD, callous-unemotional behavior, social impairment, and academic impairment), the SDQ scales (i.e., emotional problems, conduct problems, peer problems, and prosocial behavior), and the Kidscreen scales (i.e., physical health, psychological well-being, autonomy and parent relations, social support and quality of peer relations, well-being, and performance at school). For these analyses, the SCT and ADHD-IN items continued to be treated as categorical indicators with the other scales treated as manifest variables (i.e., summary scores on each scale). The purpose of these analyses was to determine the correlations of the SCT and ADHD-IN factors with the other scales.

The third series of analyses regressed the CABI, SDQ, and Kidscreen scales (see above paragraph) on the SCT and ADHD-IN factors. For these analyses as well, the SCT and ADHD-IN items continued to be treated as categorical indicators with the other scales treated as manifest variables (i.e., summary scores on each scale). The purpose of these analyses was to determine the unique associations of the SCT and ADHD-IN factors with the other scales.

The level of significance was set at alpha < 0.05. Some statistical analyses were performed with SPSS^®^, version 28.0 (IBM Corporation, Armonk, NY, USA) for Apple Mac^®^.

## 3. Results

### 3.1. Internal Validity of SCT and ADHD-IN Symptoms

An a priori SCT and ADHD-IN two factor-model with cross-loadings yielded an acceptable fit X^2^ (229) = 385, *p* < 0.001, CFI = 0.971, RMSEA = 0.057 (0.047, 0.067), and SRMR = 0.066. The loadings of the 15 SCT items on the SCT factor were moderate to substantial (M = 0.63, SD = 0.11, range = 0.43 to 0.78) with 11 of the 15 SCT items having non-significant cross-loadings on the ADHD-IN factor (M = 0.11, SD = 0.15, range = −0.08 to 0.37). Only two SCT items (*lost in a fog* and *loses train of thought*) had relatively lower loadings on the SCT factor (0.49 and 0.43, respectively) in conjunction with cross-loadings on the ADHD-IN factor of approximately 36. All SCT items thus had higher loadings on the SCT factor than the ADHD-IN factor. The nine ADHD-IN items had moderate to substantial loadings on the ADHD-IN factor (M = 0.69, SD = 0.14, range = 0.53 to 0.89). Six of the nine ADHD-IN items had non-significant cross-loadings on the SCT factor (M = 0.12, SD = 14, range = −0.06 to 0.31). Table 1 shows the loadings of the SCT and ADHD-IN items on the SCT and ADHD-IN factors.

### 3.2. Correlations of SCT and ADHD-IN Factors with CABI, SDQ, and Kidscreen Scales

Table 2 shows the correlations of the SCT and ADHD-IN factors with the other CABI scales and the SDQ and Kidscreen scales. Higher scores on the SCT and ADHD factors were associated with higher scores on the CABI anxiety, depression, ADHD-HI, ODD, callous-unemotional, social impairment, and academic impairment scales (*ps* < 0.001). Higher scores on the SCT and ADHD-IN factors were associated with higher scores on the SDQ emotional problems, conduct problems, and peer problems scales, and lower scores on the SDQ prosocial behaviors scale (*ps* < 0.001). For the Kidscreen scales, the higher scores on the SCT factor were associated with lower scores on the psychological well-being, autonomy/parent relationships, and social support/quality of peer relations (*ps* < 0.001) with higher scores on the ADHD-IN factor associated with lower scores on the psychological well-being and social support/quality of peer relations (*ps* < 0.01).

### 3.3. Unique Relationships of SCT and ADHD-IN Factors with CABI, SDQ, and Kidscreen Scales

Table 3 shows the partial standardized regression coefficients from the regression of the CABI, SDQ, and Kidscreen scales as manifest variables on the SCT and ADHD-IN factors. For the CABI scales, higher scores on the SCT factor had significant (*ps* < 0.01) unique associations with higher scores on the anxiety, depression, ADHD-HI, and ODD scales while higher scores on the SCT factor did not have significant unique associations with scores on the callous-unemotional behavior, social impairment, and academic impairment scales. In contrast, higher scores on the ADHD-IN factor had significant (*p* < 0.01) unique associations with higher scores on the depression, ADHD-HI, ODD, callous-unemotional behavior, social impairment, and academic impairment scales. The ADHD-IN factor did not have a significant unique association with the anxiety scale.

For the SDQ scales, higher scores on the SCT factor had significant (*ps* < 0.05) unique associations with higher scores on the emotional problems, conduct problems, and peer problems scales, and lower scores on the prosocial behaviors scale. In contrast, the ADHD-IN factor only had a significant (*p* < 0.05) unique association with the conduct problems scale. For the Kidscreen scales, higher scores on the SCT factor had significant (*ps* < 0.05) unique associations with lower scores on the psychological well-being, autonomy/parent relationships, and social support/quality of peer relationships scales, while the higher scores on the ADHD-IN factor only had a significant (*p* < 0.05) unique association with lower scores on the social support/quality of peer relationships scale.

## 4. Discussion

The aims of the current study were to translate the Child and Adolescent Behavior Inventory (CABI)–Parent Version [22] into Farsi, to test its psychometric characteristics, and to investigate, whether and to what extent SCT traits were associated with specific psychological functioning such as internalizing problems and social problems with parents and peers compared to ADHD-IN traits. Results showed that the psychometric properties of the Farsi version were satisfactory, that two clearly distinguishable factors of SCT and ADHD-IN traits could be identified, that SCT traits were more strongly associated with internalizing problems, that ADHD-IN traits were more strongly associated with externalizing problems, and that both SCT and ADHD-IN showed similar associations with social impairments.

Given the aforementioned, the present results add to the current literature in four important ways: First, the Farsi version of the CABI is psychometrically sound, and as such, both scientists and practitioners working in Farsi-speaking areas may benefit from this measure. Second, the pattern of results underlines the transdiagnostic and, above all, transcultural value of the SCT construct. Third, and similarly, SCT is clearly distinguishable from ADHD, in general, and from ADHD-IN, more specifically. Fourth, SCT symptoms such as mental confusion, daydreaming, lethargy, slow in completing tasks, lack of initiative and efforts, and trouble staying awake and alert have negative consequences for the individual with SCT and their social environment.

### 4.1. SCT and ADHD-IN Traits

With the first hypothesis, we assumed that SCT scores loaded on a factor clearly distinguishable from the factor ADHD-IN, and data did confirm this. Accordingly, the present results are in accord with previous studies [5,6,23]. We expanded upon previous research in that SCT traits were identified among a sample of Farsi-speaking children and adolescents, and in that we confirmed what has already been observed among children and adolescents living in the U.S., Turkey, South Korea, and Spain.

### 4.2. SCT and ADHD-IN Traits Related to Other Psychological Dimensions and Behavior

Similar to previous studies [5,6,23], SCT symptoms were more strongly correlated than ADHD-IN symptoms with internalizing symptoms such as anxiety and depression (Table 3). Thus, we confirmed the second hypothesis. In contrast, when compared to SCT symptoms, ADHD-IN symptoms were more strongly associated with ADHD-hyperactivity/impulsivity, CU traits, and social and academic impairments, when assessed with the CABI (Table 3). Thus, we also confirmed the third hypothesis. Further, and against expectations, SCT traits more significantly correlated to oppositional defiant disorder (ODD; r = 0.43), while for ADHD-IN, the magnitude of this association was lower (r = 0.28). While the quality of the data does not allow a deeper understanding of the underlying psychological mechanisms, we discuss these results in the following section.

### 4.3. SCT, ADHD-IN, Oppositional Defiant Disorder, and Social Problems

Previous research showed that SCT traits and ADHD-IN symptoms were both associated with social impairments [5,6,23], and callous-unemotional (CU) behaviors [5], while in another study [6], ADHD symptoms were more strongly associated with oppositional defiant disorder symptoms. The research question asked whether and to what extent SCT and ADHD-IN traits were associated with dimensions of conduct disorder and social issues.

Against what has been observed so far in the context of SCT traits and ODD [2,5,6,15,18,23,25,35,36], children and adolescents scoring high in SCT traits also scored high in symptoms of ODD. The quality of the data does not allow a deeper understanding of such an association. Perhaps this pattern of results is a specific cultural trait only observable so far among Iranian children and adolescents, though without being able, so far, in identifying such cultural factors. Next, given that the present data are parents’ judgements, it is further conceivable that children’s and adolescents’ demanding behavior was labeled as disobedient, and thus oppositional. It may also be that the observed association is a random result. As such, a replication study should help to shed some more light on this issue. A further explanation might be that traits of ODD belong to issues related to social behavior and conduct problems. Indeed, when assessing social behavior and conduct problems with the SDQ (Table 3), SCT traits were associated with peer problems (r = 0.25), low social behavior (r = 0.21), conduct problems (r = 0.20), and emotional problems (r = 0.39). As regards the associations between SCT traits and psychological issues related to a social context, the present results mirror the high overlap between scores for SCT and social withdrawal [37]. As such, it appears that patterns of associations between SCT and psychological issues might also differ based on the quality and type of measures. For ADHD-IN, such dimensions were associated with callous-unemotional behavior (Table 3; r = 0.26), when assessed with the CABI, with lower social support and lower peer quality relationship (Table 3; r = −0.31), when assessed with the Kidscreen, while ADHD-IN symptoms were statistically unrelated with peer problems of low prosocial behavior (Table 3), when assessed with the SDQ. Collectively, the current data provide sufficient evidence that both SCT and ADHD-IN appear to be associated with social behavioral issues, though, the pattern of results is not uniform and may vary as a function of the applied measures. As such, it appears further justified to apply a broader range of available and psychometrically well-tested measures.

### 4.4. SCT and Social Problems; Possible Explanations

Next, while the present data showed *that* SCT and social issues were associated, the quality of the data does not answer *why* this happened. Given that the present study has a cross-sectional design, by nature, causal relationships cannot be drawn. Theoretically and hypothetically, four processes are conceivable: reactive, evocative, interactional, and proactive processes.

First, an individual reporting issues with their parents and peers might withdraw from social contacts and pull back into their inner world. Thus, the individual with SCT might be in a reactive state regarding their social environment.

Second, an individual’s behavior of excessive daydreaming, mental confusion, fogginess, and slowed behavior and thinking might trigger social distancing, social rejection, and conflicts with peers and parents (see Table 2 and Table 3). Thus, the individual with SCT might be in an evocative state regarding their social environment.

The third assumption (again highly speculative and not possible to test with the current data) considers bi-directional influences over time, and thus an interactional process: To this end, and for want of similar models for children and adolescents with SCT, we rely on three models: First, Patterson et al.’s model for the development and maintenance of the children’s coercive behavior [38] model claims that adverse parenting styles such as high behavioral and responsive inconsistency, low control and lack of warmth interact with a child’s unfavorable temperament (e.g., easily irritable, irascible, low tolerance of frustration), behavior (e.g., oppositional-aggressive, hyperactive), and intellectual skills (e.g., low degree of fast and accurate information processing) and vice versa. Thus, this model claims a reciprocal impact and feedback loop between a parent’s problematic parenting style and a child’s psychosocial, intellectual, and behavioral characteristics over time. Second, and from a psychological and developmental area other than SCT: When an eight-year-old child reported that their father was absent, that their mother was less sensitive in interacting with their child, and that the mother–child relationship was poor, this child scored high in sleep problems three years later at the age of 11 years. Next, a child’s sleep problems at their age of 11 years predicted adverse changes in maternal emotionality and sensitivity, when the child crossed into adolescence at the age of 14 three years later [39]. Third, Sameroff’s interactional model [40,41] underscores the importance of family functioning and family and peer interaction patterns in the emergence and maintenance of a child’s problematic and dysfunctional behavior. Likewise, we may speculate that, over time, a child or adolescent with SCT traits and the context of unfavorable and conflicting social interactions may reciprocally influence one another, as the present data seem to indicate. In sum, one may claim that both individuals high in SCT and their social environment may be ‘‘swept’’ into a sort of ‘‘negative spiral’’ in which reciprocal effects occur. As such, following Sameroff’s interactional model of the emergence and maintenance of a child’s disruptive and externalizing behavior [40,41], we might also speculate that the interventions on family level might mitigate a child’s SCT traits.

The fourth assumption highlights a person’s proactive interaction. As children grow older, they move beyond the social environment provided by their parents and begin to select and construct their own social environment. As such, it is conceivable that children and adolescents with SCT choose their peers, who better match with their SCT traits. In sum, while we acknowledge that these theoretical assumptions are clearly hypothetical, future and longitudinal studies should have the power to shed some more light on such reciprocal processes.

### 4.5. Strengths and Limitations of the Study

The strengths of the present study are the large sample size, and the translation of the measure with low disagreement.

In addition, the pattern of results should be balanced against the following limitations: First, here, we exclusively reported the psychometric validation of the Parent Version.

Second, latent and unassessed confounders might have biased the pattern of results. To illustrate, higher SCT traits were associated with impaired sleep [42]. As such, future studies on SCT might introduce subjective and objective sleep assessments, along with dysfunctional pre-sleep cognitive-emotional processes such as bed-time and sleep-time procrastination [43,44,45,46,47,48,49] to identify, if and to what extent poor sleep and pre-sleep-related cognitions might both trigger and maintain higher symptoms of SCT [50]. In a similar vein, the associations between patterns of regular subjectively perceived and objectively measured physical activity and exercising on dimensions of SCT traits have not been investigated so far. In contrast, as regards ADHD, it appears that regular physical activity and exercising have the potential to favorably impact on attention and executive functions [51] and hyperactivity [52]. Given this, future studies should carefully assess, if and if so, to what extent regular physical activity and exercising have the potential to favorably impact on SCT traits.

Third, and relatedly, there are some sparse, though encouraging observational and interventional studies, which showed that increased engagement in physical activity and exercising might improve dimensions of social contacts, at least among adults with psychiatric issues [53], and emotional competencies, at least among females with multiple sclerosis [54]. As such, and given that social withdrawal is a serious issue in individuals with SCT [37], it is conceivable that regular physical activity and exercising might favorably impact on dimensions of social behavior also among individuals with SCT.

Fourth, interestingly, and unexpectedly, higher SCT and ADHD-IN scores were associated with higher physical health and school environment. The current literature does not provide a reasonable explanation for such a pattern of results. Accordingly, we have re-run all statistical computations once again, and we have carefully checked, if mistakenly, we have misunderstood the anchor points of the questionnaires, though, we could not identify any mistakes. Given this, we were unable to provide any reasonable explanation. We also note that all questionnaires were psychometrically sound. Though, and against expectations, in the present sample, higher scores for SCT were also associated with higher scores for ODD; as such, it appears conceivable that perhaps latent and unassessed (cultural) confounders might have biased the present data in the same or opposite directions. Clearly, the present and unexpected findings need to be replicated in the future.

Fourth, given that the pattern of results showed that children and adolescents scoring high on SCT and ADHD-IN, as rated by their parents, also scored high on social issues, future studies should assess the quality of peer and family relationships and interactions more accurately.

Fifth and last, data from representative samples of youth showed that excessive use of screen media and social network sites (SNSs) might be problematic [55,56,57,58]. Given this background, children and adolescents with SCT should be routinely asked about their screen media habits.

## 5. Conclusions

The Farsi version of the CABI–Parent Version has sufficient and satisfactory psychometric properties to accurately identify children and adolescents with SCT, in contrast to those with ADHD-IN. Further, SCT is associated with a broad variety of internalizing and social issues. Lastly, SCT appears to be a transdiagnostic and transcultural psychiatric issue.

## Figures and Tables

**Table 1 jcm-11-06346-t001:** Standardized Primary and Secondary Factor Loadings (standard errors) of Sluggish Cognitive Tempo (SCT) and ADHD Inattention (IN) Symptoms on Sluggish Cognitive Tempo and ADHD-Inattention Factors.

Sluggish Cognitive Tempo Symptoms	SCT Factor	ADHD-IN Factor
1. Behavior is slow (sluggish)	0.52 (0.10)	0.13 (0.11) ^ns^
2. Lost in a fog	0.49 (0.10)	0.37 (0.10)
3. Stares blankly into space	0.72 (0.10)	−0.03 (0.11) ^ns^
4. Drowsy or sleepy (yawns) during the day	0.76 (0.06)	−0.08 (0.08) ^ns^
5. Daydreams	0.64 (0.06)	0.00 (0.06) ^ns^
6. Loses train of thought	0.43 (0.09)	0.36 (0.08)
7. Low level of activity (underactive)	0.64 (0.06)	0.01 (0.07) ^ns^
8. Gets lost in own thoughts	0.56 (0.09)	0.30 (0.09)
9. Easily tired or fatigued	0.78 (0.06)	−0.05 (0.08) ^ns^
10. Forgets what is going to say	0.60 (0.08)	0.00 (0.10) ^ns^
11. Easily confused	0.61 (0.07)	0.27 (0.08)
12. Spaces or zones out	0.55 (0.10)	0.08 (0.11) ^ns^
13. Gets mixed up	0.72 (0.06)	0.00 (0.07) ^ns^
14. Thinking is slow	0.76 (0.07)	0.10 (0.09) ^ns^
15. Difficulty expressing thoughts	0.60 (0.07)	0.16 (0.08) ^ns^
**ADHD-Inattention Symptoms**		
1. Close attention to detail	−0.04 (0.09) ^ns^	0.82 (0.07)
2. Sustaining attention	0.00 (0.01) ^ns^	0.87 (0.04)
3. Listening when spoken to directly	0.06 (0.09) ^ns^	0.74 (0.08)
4. Follow through on instructions	−0.06 (0.10) ^ns^	0.89 (0.08)
5. Organization skills	0.25 (0.09)	0.60 (0.08)
6. Avoids tasks requiring sustained effort	0.26 (0.09)	0.53 (0.08)
7. Loses things	0.13 (0.11) ^ns^	0.55 (0.10)
8. Easily distracted	0.18 (0.10) ^ns^	0.69 (0.08)
9. Forgetful in daily activities	0.31 (0.09)	0.55 (0.08)

Note. *n* = 209. The correlation between the SCT and ADHD-IN factor was 0.63 (*SE* = 0.06). All loadings were significant at *p* < 0.05 unless noted as non-significant (ns). The complete wording of the ADHD-IN symptoms is available from the first author.

**Table 2 jcm-11-06346-t002:** Correlations (Standard Errors) of Sluggish Cognitive Tempo (SCT) and ADHD-Inattention (IN) Factors with CABI, SDQ, and Kidscreen Scales as Manifest Variables.

External Correlates	SCT Factor	ADHD-IN Factor
**CABI Scales**		
Anxiety	0.56 (0.04)	0.42 (0.05)
Depression	0.58 (0.04)	0.49 (0.04)
ADHD hyperactivity-impulsivity	0.51(0.04)	0.59(0.04)
Oppositional defiant disorder	0.61 (0.03)	0.56 (0.04)
Callous-unemotional behavior	0.18 (0.07)	0.27 (0.07)
Social impairment	0.29 (0.06)	0.37 (0.07)
Academic impairment	0.39 (0.07)	0.52 (0.07)
**SDQ Scales**		
Emotional problems	0.48 (0.05)	0.39 (0.06)
Conduct problems	0.39 (0.05)	0.42 (0.05)
Peer problems	0.29 (0.06)	0.23 (0.06)
Prosocial behavior	−0.31 (0.07)	−0.29 (0.07)
**Kidscreen Scales**		
Physical well-being	0.14 (0.07) ^ns^	0.15 (0.07) ^ns^
Psychological well-being	−0.35 (0.06)	−0.28 (0.07)
Autonomy and parent relationships	−0.23 (0.07)	−0.13 (0.07) ^ns^
Quality of social support/peer relationships	−0.47 (0.05)	−0.48 (0.06)
Well-being/school performance	0.20 (0.06)	0.24 (0.07)

*Note*. *n* = 209. All correlations are significant at *p* < 0.05 unless noted as non-significant (ns). CABI = Child and Adolescent Behavior Inventory; SDQ = Strengths and Difficulties Questionnaire. Kidscreen = Quality of Life Scale.

**Table 3 jcm-11-06346-t003:** Partial Standardized Regression Coefficients (Standard Errors) for the Association of Sluggish Cognitive Tempo (SCT) and ADHD-Inattention (IN) Factors with CABI, SDQ, and Kidscreen Scales as Manifest Variables.

External Correlates	SCT Factor	ADHD-IN Factor
**CABI Scales**		
Anxiety	0.49 (0.08)	0.10 (0.09) ^ns^
Depression	0.45 (0.06)	0.20 (0.07)
ADHD hyperactivity-impulsivity	0.24 (0.08)	0.44 (0.08)
Oppositional defiant disorder	0.43 (0.05)	0.28 (0.06)
Callous-unemotional behavior	0.02 (0.10) ^ns^	0.26 (0.10)
Social impairment	0.09 (0.09) ^ns^	0.32 (0.09)
Academic impairment	0.11 (0.09) ^ns^	0.45 (0.08)
**SDQ Scales**		
Emotional problems	0.39 (0.08)	0.14 (0.09) ^ns^
Conduct problems	0.20 (0.07)	0.29 (0.07)
Peer problems	0.25 (0.09)	0.07 (0.09) ^ns^
Prosocial behavior	−0.21 (0.10)	−0.16 (0.10) ^ns^
**Kidscreen Scales**		
Physical well-being	0.07 (0.11) ^ns^	0.11 (0.10) ^ns^
Psychological well-being	−0.29 (0.11)	−0.09 (0.12) ^ns^
Autonomy and parent relationships	−0.24 (0.10)	0.02 (0.10) ^ns^
Quality of social support/peer relationships	−0.27 (0.08)	−0.31 (0.09)
Well-being/school performance	0.09 (0.09) ^ns^	0.19 (0.10) ^ns^

*Note. n* = 209. All correlations are significant at *p* < 0.05 unless noted as non-significant (ns). CABI = Child and Adolescent Behavior Inventory; SDQ = Strengths and Difficulties Questionnaire. Kidscreen = Quality of Life Scale.

## Data Availability

Data are made available to experts in the field upon request. The request should clearly state, for which reasons data are requested, and which hypotheses should be testes.

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
