# Peer review of "Sluggish Cognitive Tempo among Iranian Children and Adolescents: A Validation Study of the Farsi Child and Adolescent Behavior Inventory (CABI)–Parent Version"

_jcm, 2022, doi:10.3390/jcm11216346_

Round 1

Reviewer 1 Report

The current paper seeks to validate a Farsi version of the CABI for parents. Cross cultural research such as this is very important so we can better understand how psychological constructs may differ in non-western cultures and so we can be sure that when we adapt measures, they are relevant and useful in other cultures. The introduction is well written, covers the reasons why investigating sluggish cognitive tempo is important, and covers other validations studies of the CABI in different languages. The methodology of this study is solid; though the reporting of methods and statistical analyses could be improved.

Introduction

-          You mention countries such as United States, Canada, Turkey, Iraq, Germany etc as countries with Farsi speaking individuals. It would be interesting to note what percentage of these populations speak Farsi, to highlight the importance of creating a measure in Farsi. Please make sure to cite where you get the rates of Farsi speaking individuals in these populations.

Methods

-          When describing the SDQ and QOL measures, please briefly describe the validity and reliability of the questionnaires.

-          In the statistical analysis section, please report what kind of factor analysis was used (eg exploratory, confirmatory, ESEM), the estimator, and rotation. Please state what criteria is used to determine if an item loads onto a factor, or whether it is crossloading on both factors.

-          Please report whether any variables or items were skewed or had non-normal distributions, and whether assumptions were met to conduct the analyses in this study.

-          For the procedure, other steps such as surveying of teachers and children are reported in detail, and probably could be more brief since this data is not investigated in the current paper.

Results

-          Item 4 of the SCT seems to cross load very badly and may be a candidate for removal, unless this is a typo. If it is a typo, please proofread all statistics in this paper to make sure it has all been written accurately.

-          The ‘unique relationships’ analyses (eg in table 3) might be better done with partial correlations coefficients rather than multiple regression. If you report regression models, then it would make more sense to report things like total variance explained etc, which you do not do.

-          Please report the correlation between SCT and ADHD-IN

-          200 parents are reported to have completed questionnaires on 373 students. But in the tables in the results, N = 200 for all of these correlation analyses. This suggests that a separate questionnaire was done for each child, and one questionnaire per parent then chosen for the analyses. Could you please clarify this discrepancy?

-          Please report inter-item reliability for the CABI subscales, and also means and SDs of each subscale.

-          Discriminant validity and convergent validity are not mentioned in this section, and should be noted when analyses demonstrate these types of validity.

-          Please report what percentage of the sample had ADHD or SCT symptoms that were over clinical cut-offs.

Discussion

-          The interpretation of results is written well, but I am curious if you think some variation in the results could be due to cross-cultural differences when compared to Asian and western cultures? Perhaps it would make more sense to write more to highlight the similarities /differences between your results and the CABI validation study in Turkish as this is the closest culture.

-          For the first paragraph of section 4.4, I would not say that any of these points are actually strengths of this study. Rather, I think the strengths are that sample is large and the translation of the measure was done with little disagreement. I don’t think you can say the results are robust because you have not repeated this study. This is the first time you have done this study on this new questionnaire. You could say that the pattern of results are robust if you compared the teacher ratings to the parent ratings, but that is not what you have written about in this paper. Points 4 and 5 in this paragraph are not strengths of the study but rather findings of the study.

-          I do not understand this sentence “Clearly, results should be further aligned to those based on self-reports.” And how this first point is a limitation of this study. Please elaborate further.

-          Are there any limitations (and future directions) or psychological constructs that were not measured that are specific to the Iranian culture?

Minor comments

-          In the second sentence of the abstract, “opposite defiant disorder” should be changed to “oppositional defiant disorder (ODD) symptoms”

-          In the first sentence in the third paragraph of the introduction is phrased strangely. Perhaps change the sentence to “For practitioners, the concept of SCT is important to further distinguish attention problems, such as daytime sleepiness and internalizing problems, from typical ADHD.”

-          Please explain what is meant by ‘SEs’ – if this is supposed to refer to Standard errors, then I cannot see them in Table 1.

-          The percentage of girls/boys differs here and in the methods section (ie 53.9% vs 54.7); please correct this.

-          In section 4.1, “proved” should be changed to another word, as it is not appropriate for scientific writing.

-          In section 4.2, “loaded” is used incorrectly as it refers to factor analysis. This word should be changed to “correlates” or “relates” or something similar.

-          There are some places in the manuscript where the language is colloquial. Please revise the language used in this paper so it is more formal.

Author Response

We thank Reviewer #1 for their valuable comments, which helped us to improve the quality of the manuscript. Please find the detailed point-by-point-response attached as a separate file. 

Once again, thank you so much for all your kind efforts.

Reviewer 2 Report

Thank you for the opportunity to review this paper that addresses the novel neurodevelopmental profile of Sluggish Cognitive Tempo (SCT) and explores the validity of the Child and Adolescent Behavior Inventory (CABI)-Parent Version in Farsi for distinguishing SCT from other related childhood conditions within the Farsi speaking population. I believe this paper would be of interest to practicing physicians who may wish to determine whether their patients or clients have this profile and would like to know whether the Farsi version of the Child Behavior Inventory can provide a valid measure of this profile type that can aid in clinical identification.

The title as written did not fully address the core content of the paper. The manuscript as written is principally about Sluggish Cognitive Tempo (SCT) but there is no mention of this in the title.  

Introduction

The opening paragraphs frames the paper in terms of the importance of clinical interest in SCT and its potential need for distinct management from ADHD.  I would like to see just a brief mention of the potential here to “over-pathologize” something that might at times be experienced by a significant proportion of the population.  Some of the benefit of identifying SCT might be in the recognition that this represents another aspect of neurodiversity and that individuals with this picture may need different adaptations in their environments, and increased understanding from families, teachers and peers, in addition to any consideration of need for medication.

In Section 1.1. the authors report on prior use of the CABI to identify children with SCT in different populations including US, South Korea, Turkey and Spain.  In paragraph 2 of this section the authors list findings from a study on 2,056 mothers of children who completed the CABI.  It is not stated which country these mothers were from – can the authors add that this is the US group? Reference 6 clarifies that scores greater than the top 5% on SCT and ADHD measures were used to create SCT-only (n = 53, 2.58%), ADHD-only (n = 93, 4.52%), SCT+ADHD (n = 49, 2.38%), and comparison (n = 1,861, 90.52%) groups.  This information was very helpful for me in understanding how the authors were distinguishing between these different groups, and I would recommend that this clarification is included in the text. 

The final paragraph of section 1.1 refers to two studies using the Spanish version of the CABI.  One of these studies addresses symptoms of psychopathology of children in foster care compared with those not in foster care and is the one described in the text.  The other study (reference 24) entitled “Optimal Items for addressing SCT in children across Mother, Father and Teacher ratings” reports on the CABI items that are used to distinguish SCT from ADHD-IN in a Spanish speaking population.  This study seems more relevant to the main focus of this paper than the foster care paper – could the authors clarify? 

The introduction as written suggests some lack of clarity as to whether the main aim of this study is to test all of the psychometric properties of the Farsi version of the CABI; to determine whether the Farsi version can distinguish between SCT alone, ADHD alone and SCT +ADHD as has been shown in the other language group studies, or to describe the different patterns of association noted between SCT groups across different cultures. This point can readily be clarified by the authors.

Section 1.2 describes the current study.   The second paragraph states “The three hypotheses and research questions were as follows: First, based on previous studies, we assumed the SCT scores loaded on a factor clearly distinguishable from the factor ADHD-IN.”  I am unclear as to the meaning-is there a word missing?  Are the authors reporting the way in which they combine items into factor scores – could they add some clarifying language with some more detail for clinicians who may be less familiar with factor loading and factor analysis? Should this read “…were clearly distinguishable from.  “Similarly, the research question given at the end of the second paragraph could benefit from some clarification e.g. “We sought to determine whether, in the Farsi population, SCT symptom dimensions identified through use of the CABI differed in their patterns of association with other psychopathology and with impaired social behavior when compared with ADHD-IN symptom dimensions” (or similar language).

Methods

In the procedure section the authors state that schools were identified following clustered sampling – what criteria were used to divide the schools into clusters- gender first, then age? Others? How many schools were approached from each cluster and how many agreed to participate?  Similarly, how many parents were invited to participate in the study, how many of those that signed the consent, and how many completed the CABI i.e., what was the response rate?  Were respondents skewed in any way?   This gives some idea of the representativeness of the parent sample and hence the generalizability of the results to the general population. It also gives some idea of whether the CABI completion was acceptable to this population.

Results

The results are clearly presented and in general show good discriminant validity for the Farsi version of the CABI between SCT Items and ADHD-IN items as compared in a two-factor model.  SCT and ADHD-IN also show different patterns of association with other psychopathology e.g., anxiety, depression and ODD.  Interestingly (line 328) the authors state that higher SCT and ADHD-IN scores were associated with higher physical health and school environment- I don’t think this was addressed in the discussion.  Would the authors wish to discuss possible reasons for this?  

Discussion

Although this section references relevant literature, given the likely readership of the journal it could be made more relevant for clinicians.  I appreciate the authors desire not to move too far into speculation, but I think some of the implications of their findings for future research could be explored more fully.  One aspect that may be worthy of some deeper discussion is the potential mechanisms underlying the types of association observed in the study.  For example, that alluded to in line 412 “An individual reporting interactional issues with their parents and peers might withdraw from social contacts and pull back in their inner world.”  This might represent a kind of “reverse causality” where external factors are triggering the behavior described as SCT rather than SCT representing an intrinsic biological difference.  This point is very important because if the root cause of the behavior lies, say in early family interaction difficulties, or even in exposure of the child to traumatic early experiences (also known to be associated with withdrawal behaviors), then the solutions likely do not lie in the prescription of medication to the child.  Could the authors expand on this thinking a little further and perhaps look more widely in the literature for associations with family issues and trauma?  Perhaps also in considerations of next seps for research, there could be a recommendation for wider study of environmental factors, including family function and family and peer interaction patterns (not currently widely evaluated) in relation to SCT and potentially also explore opportunities for family-level interventions? The discussion of bi-directional influences is good but could draw much more widely from the child development literature including Sameroff’s transactional model and more recent life course  models that incorporate the influence of prior events and experiences in addition to aspects of the family, social and physical environment on aspects of health status.  

Author Response

We thank Reviewer #2 for their valuable comments, which helped us to improve the quality of the manuscript. Please find the detailed point-by-point-response attached as a separate file.

Thank you once again for all your kind efforts.

Round 2

Reviewer 1 Report

The manuscript looks like it is improved a lot. Thank you for taking my comments on board.